# Dynamic Training Guided by Training Dynamics

## Abstract

This paper centers around a novel concept proposed recently by researchers from the control community where the training process of a deep neural network can be considered a nonlinear dynamical system acting upon the high-dimensional weight space. Koopman operator theory, a data-driven dynamical system analysis framework, can then be deployed to discover the otherwise non-intuitive training dynamics. Different from existing approaches that mainly take advantage of the prediction capability of this framework, we take a deep dive into understanding the underlying relationship between the low-dimensional Koopman modes that describe the training dynamics and the weight evolution itself, and develop two novel strategies for speeding up model convergence in an online fashion. These include 1) a gradient acceleration strategy that improves training efficiency by pushing the slowly decaying Koopman modes to decay faster, and 2) a masking strategy that drastically reduces the computational complexity of gradient acceleration by analyzing the contribution of the corresponding Koopman modes in weight reconstruction. These strategies offer promising insights into pursuing faster and more efficient training methodologies and improve our understanding of training dynamics to further control and inform the training process.

## 1 Introduction

The advent of cutting-edge hardware (Li et al., 2014) and the development of parallel processing techniques (Li et al., 2020) have greatly accelerated the training process of Deep Neural Networks (DNN). However, enhancing the fundamental techniques of DNN training continues to be a significant challenge. From the inception of Stochastic Gradient Descent (SGD) (Robbins & Monro, 1951), which has since become a mainstay in DNN training, numerous techniques have been proposed to increase the efficiency of the underlying optimization task. One major limitation of SGD is its sensitivity to hyperparameters, particularly learning rates.

Various methods have been developed to improve SGD's performance in this area, such as learning rate annealing and momentum (Sutskever et al., 2013), providing robustness against local minima and regions with low derivatives. Several adaptive optimization algorithms have also been proposed. Notably, the Adam optimizer (Kingma & Ba, 2014) has been a significant advancement, which leverages the first and second moments of gradients to adaptively adjust the learning rate for each weight. Besides these first-order optimizers, attention has also been given to second-order optimizers (Martens, 2010) that use the curvature information or second-order derivatives of the loss function to guide the optimization process, potentially enabling a more efficient convergence.

A host of auxiliary strategies have also been developed to further improve the training efficiency. For instance, batch normalization (Ioffe & Szegedy, 2015) standardizes the input to each layer, accelerating training and reducing the model's sensitivity to initialization; early stopping (Prechelt, 2002) halts the training process when the model's performance on a validation set begins to decline, thereby preventing overfitting and enhancing generalization; and the freezing methods (Yang et al., 2023; Yuan et al., 2022; Liu et al., 2021; Yuan et al., 2022) obviate the need to compute the gradients of the frozen parameters, providing improved computational efficiency and overfitting prevention.

Very recently, a novel interpretation of the DNN training process has been proposed, mainly by researchers from the control community (Redman et al., 2022; Dogra & Redman, 2020; Manojlovic

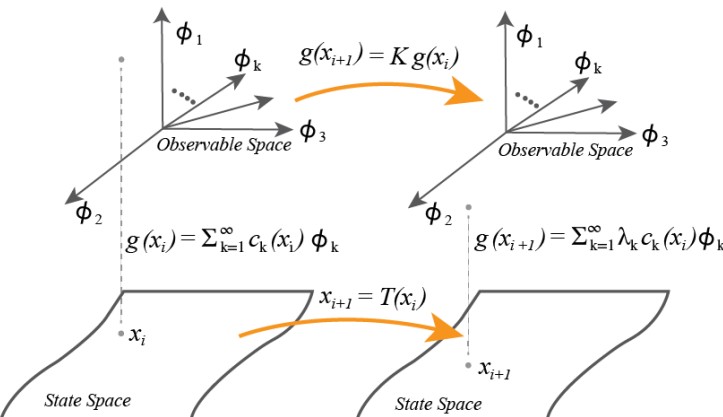

Figure 1: Illustration of Koopman state space. In state space (bottom) the nonlinear function $T$ maps the state $x_i$ to $x_{i+1}$. The Koopman operator, $K$, acts on the space of observables (top) characterizing the 'dynamics of observables'.

et al., 2020; Tano et al., 2020) – If it is intuitive to consider a pre-trained DNN as an inherently nonlinear static system acting upon the high-dimensional inputs, then **the DNN training process itself is essentially a nonlinear dynamical system acting upon the high-dimensional weight space**! It is a discrete dynamical system since the weights of a DNN evolve over each epoch according to the optimization process adopted. This drastically different interpretation has led to the establishment of a novel mathematical framework for learning, where Koopman operator theory (Mezić, 2005), a powerful data-driven dynamical system analysis tool, is adopted to exploit the underlying dynamics in the training process of a DNN. This also brings huge benefit that we could take advantage of recent advances in data-driven dynamical system analysis for an in-depth investigation of the seemingly non-intuitive training process.

This paper builds upon these recent developments but focuses specifically on how to utilize the underlying training dynamics discovered using the Koopman Operator Theory to help speed up this dynamic training process. In the following, we first provide brief background material on Koopman operator theory.

## 2 BACKGROUND AND RELATED WORKS

Since the paper centers around the interpretation of DNN training process as a nonlinear dynamical system, this review is tailored toward related works in dynamical system analysis.

A wide variety of dynamical system analysis approaches have been developed in recent years, including the projection-based methods (Benner et al., 2015; Holmes et al., 1996; Towne et al., 2018; Berkooz et al., 1993) and decomposition-based methods (Schmid, 2010; Rowley et al., 2009; Kutz et al., 2016; Mezić, 2005). Machine learning approaches have also been employed to select nonlinear functions from a large library that best capture the temporal evolution of observables (Mangan et al., 2019; Brunton et al., 2016; Pantazis & I. Tsamardinos, 2019). However, majority of the machine learning approaches are only suitable for analyzing dynamics represented by a relatively simple and low order set of nonlinear differential equations.

The key notion of Koopman analysis is the representation of a (possibly nonlinear) dynamical system as a linear operator on a typically infinite-dimensional space of functions (Mezić, 2021; 2005; Mezić & Banaszuk, 2004). Koopman-based approaches directly contrast with standard linearization techniques that consider the dynamics in a close neighborhood of some nominal solution. Indeed, Koopman analysis can yield linear operators that accurately capture fundamentally nonlinear dynamics.

**Koopman Operator Theory.** As a brief description, consider a discrete-time dynamical system $x_{i+1} = T(x_i)$, where $x_i \in \mathbb{R}^n$ is the current state and $x_{i+1}$ is the next state after application of

the potentially nonlinear mapping $T$. Consider also a vector-valued observable $g(x) \in \mathbb{R}^m$. The evolution of observables under this mapping can be described according to

$$g(x_{i+1}) = g(T(x_i)) = Kg(x_i). \tag{1}$$

where $K$ operates on the vector space of observables and maps $g(x_i)$ to $g(x_{i+1})$ and is referred to as the "Koopman operator" associated with the fully nonlinear dynamical system (see Fig. 1).

The Koopman operator is linear, following from linearity of the composition operator, but also infinite-dimensional. As such, for dynamical systems with a pure point spectrum for observables (Mezić, 2020), its action can be decomposed according to

$$g(x_{i+1}) = Kg(x_i) = \sum_{k=1}^{\infty} \lambda_k^{i+1} c_k(x_0)\phi_k, \tag{2}$$

where $\lambda_k$ is an eigenvalue associated with the "Koopman mode" or eigenfunction, $\phi_k$, and $c_k(x)$ is the reconstruction coefficient associated with projecting $g$ onto the eigenspace. It immediately follows that

$$g(x_{i+\alpha}) = \sum_{k=1}^{\infty} \lambda_k^{\alpha} c_k(x_i)\phi_k \tag{3}$$

for any $\alpha \in \mathbb{N}$. Eq. 3 provides a convenient and general framework to predict and control a given dynamical system. Each Koopman mode evolves over time with its frequency and decay rate governed by the argument of $\lambda_k$ and $|\lambda_k|$, respectively.

Koopman-based techniques are particularly useful in a data-driven setting because they only require measurements of observables. As such, they can be implemented even when the underlying model dynamics are unknown.

**Dynamic Mode Decomposition (DMD).** When using Koopman-based approaches, it is critical to identify a suitable *finite* basis with which to represent the infinite dimensional Koopman operator. Dynamic Mode Decomposition (DMD) (Schmid, 2010) is one standard approach for inferring Koopman-based models. It uses least-squares fitting techniques to approximate a finite-dimensional linear matrix operator, $A$, that advances high-dimensional measurements of a system forward in time:

$$g(x_{i+1}) = Ag(x_i) \tag{4}$$

where $A$ is an approximation of the Koopman operator, $K$ in Eq. 1 restricted to a measurement subspace spanned by direct measurements of the state $x$. Since the weight space of a neural network is a *fully observable* system, we define $g(x)$ to be the identity function in this work. In practice, we often use "snapshots" of the system arranged into two data matrices, $X_i$ and $X_{i+1}$, where columns of these two matrices indicate measurements taken at a certain time, and $X_{i+1}$ is $X_i$ shifted by one time step. Hence,

$$X_{i+1} \approx AX_i, \tag{5}$$

and $A$ can be solved by

$$A = X_{i+1}X_i^+ = X_{i+1}V\Sigma^{-1}U^T \tag{6}$$

where $X_i = U\Sigma V^T$ is the Singular Value Decomposition (SVD), and $X_i^+$ denotes the pseudo-inverse of $X_i$. A comprehensive discussion of DMD has been provided in (Kutz et al., 2016).

**DNN Training as a Dynamical System.** There have been a few works in recent years that adopt Koopman-based approaches to accelerate the training process of a general-purpose DNN model (Dogra & Redman, 2020; Tano et al., 2020; Manojlovic et al., 2020), although Dietrich et al. (2020) is generally considered the first work that establishes the connection between Koopman operator theory and acceleration of numerical computation. Dogra (2020) is also one of the pioneer works but with a focus specifically on neural networks for solving differential equations. Generally speaking, these works take advantage of the prediction capability of the Koopman operator theory framework, as shown in Eq. 3, to directly predict network weights a few epochs later, thus bypassing the time-consuming SGD iterations. However, we show that these methods are not suitable for obtaining an optima comparable to that obtained by traditional optimization techniques, especially when the network size is large.

**Contributions.** Different from the above prediction-based acceleration, we take a deep dive into the dynamics extracted from Koopman analysis and take advantage of the rich information contained in the Koopman operator. Specifically, we make the following two important contributions:

(1) Koopman-based Gradient Acceleration (KGA): In Koopman mode decomposition (Eq. 3), the modes that decay slowly usually lead to good representation of dynamics as they are less sensitive to noise. Nonetheless, these modes also inhibit convergence. We develop an online gradient acceleration strategy that speeds up convergence by pushing the slowly decaying modes to decay faster.

(2) Koopman-based Weight-level Masking (KWM): The Koopman mode decomposition, especially the eigenvalues (Eq. 3), also indicate the contribution of each mode in reconstructing the measurement (i.e., the network weight). We thus construct a mask such that DMD only applies to a subset of weights with significant contributions from the slowly decaying modes. This mask effectively improves the computation efficiency of KGA while preserving the rate of acceleration.

## 3 METHODS

Our goal is to improve the efficiency of the training process in an "online" fashion, taking advantage of the extracted training dynamics from data-driven dynamical system analysis, e.g., DMD. We discover the intrinsic correlation between the Koopman mode and convergence trend of each and every weight, based on which we develop two techniques, Koopman-based gradient acceleration (KGA) and Koopman-based weight-level masking (KWM), for not only faster convergence but also better optima. In the following, we first discuss the different representations of training dynamics.

### 3.1 REPRESENTING TRAINING DYNAMICS

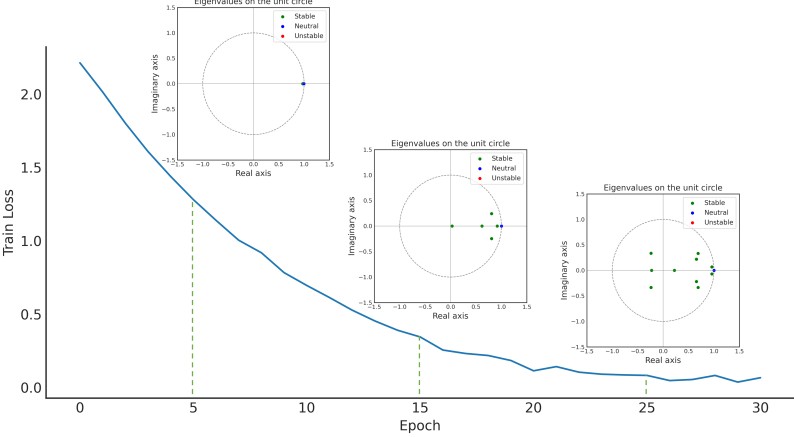

Figure 2: The correlation between training loss (that reflects the evolution of the network weights) and the eigenvalues (shown in unit circle that reflects the evolution of Koopman modes). Note: the loss curve is to train AlexNet using CiFAR10 with SGD.

Fig. 2 illustrates, at a high level, the correlation between training loss (that reflects the weight evolution) and the eigenvalues ($\lambda_k$ in Eq. 2 that reflects the evolution of the Koopman modes). Following the convention in (Avila & Mezić, 2020), we divide the eigenvalues and the associated Koopman modes into three groups, including 1) the **neutral modes** with eigenvalues close to the unit circle, i.e., $1 - \epsilon \leq |\lambda_k| \leq 1 + \epsilon$, with $\epsilon$ being a very small value; 2) the **stable modes** with eigenvalues within the unit circle, i.e., $|\lambda_k| < 1 - \epsilon$; and 3) the **unstable modes** with eigenvalues that fall outside the unit circle, i.e., $|\lambda_k| > 1 + \epsilon$. Stable modes are those that slowly decay over time, while unstable modes grow over time, thus indicating instability. The neutral modes are usually responsible for periodic behavior of the system that may exhibit recurrent patterns.

## 3.2 KOOPMAN-BASED GRADIENT ACCELERATION (KGA)

The key motivation behind KGA is to leverage the "stable Koopman modes" to selectively acceler­ate convergence along the slowest-moving directions. The stable modes capture persistent dynamics governing long-term training trajectories. KGA projects the *gradient update* onto these stable eigen­vectors, then amplifies the aligned components to speed up gradient descent along stable Koopman eigenmodes. Rather than uniformly boosting the entire gradient by changing the learning rate, as in SGD or Adam, KGA takes advantage of the extracted training dynamics and uses it to guide accelerated optimization.

By interpreting the training process as a dynamical system acting upon the weight space, the mea­surements of the system state are then the network weight after each iteration or epoch. In the context of DMD, denote the snapshot (Eq. 5) of this dynamical training process as $W_{i+1}$ and $W_i$, where $W \in R^{m \times t}$ is the matrix containing model weights over the last $t$ epochs, and $m$ indicates the number of parameters in the network. Again, $W_{i+1}$ is $W_i$ shifted by one time step. Due to the high dimension of the weight space, instead of adopting the regular SVD, a reduced SVD (Donoho & Gavish, 2013) is often used to approximate $W_i \approx U\Sigma V^T$, where we utilize only the top $r$ singular values, $\Sigma \in R^{r \times r}$, and their corresponding singular vectors, $U \in R^{m \times r}$ and $V \in R^{t \times r}$, to calculate the $r \times r$ projection of $A$,

$$\tilde{A}_{r \times r} = U^T W_{i+1} V\Sigma^{-1} \tag{7}$$

and $r$ is determined by the method described by (Donoho & Gavish, 2013). Intuitively, $\tilde{A}$ depicts the training dynamics that approximates the Koopman operator when working in a reduced order coordinate basis defined by the SVD.

In practice, however, the size of $W_i$ can be prohibitively large, especially for large models. Instead of storing $W_i$ directly, we propose to store $U$ and the projection of $W_i$ onto $U$, i.e., $\tilde{W}_i = U^T W_i$. We refer to this variant of KGA the "Efficient-KGA" for storage efficiency purpose. $\tilde{A}$ in Eq. 7 can then be calculated by

$$\tilde{A}_{r \times r} = \tilde{W}_{i+1} V\Sigma^{-1}. \tag{8}$$

Performing the eigendecomposition on $\tilde{A}$, we obtain

$$\tilde{A}Q = Q\Lambda, \tag{9}$$

where the columns of $Q$ contain the eigenvectors and $\Lambda$ is a square matrix with associated eigenval­ues on the diagonal.

To accelerate convergence, the gradient $\Delta w = w_{i+1} - w_i$ calculated by the optimizer is first pro­jected onto the eigenspace using the left singular vectors

$$\Delta\tilde{w} = U^T\Delta w \tag{10}$$

We then project the weight update to the space spanned by only the stable eigenvectors to obtain the slowly moving components,

$$\Delta\hat{w} = L_{\text{stable}}\Delta\tilde{w} \tag{11}$$

where $L_{\text{stable}}$ is a matrix containing only the stable left-eigenvectors of $\tilde{A}$. These components are then scaled by their respective eigenvalue in the diagonal $\Lambda_{\text{stable}}$ matrix and projected back to the reduced space using the stable right-eigenvectors, $Q_{\text{stable}}$. This acceleration vector is multiplied by a scalar $\alpha$, referred to as the "acceleration rate", and then added to the original change from the optimizer. The effect of $\alpha$ is investigated in Sec. 4.

$$\Delta\tilde{w} = \Delta\tilde{w} + \alpha \left( Q_{\text{stable}}\Lambda_{\text{stable}}\Delta\hat{w} \right) \tag{12}$$

$\Delta\tilde{w}$ is then projected back to the weight space by

$$\Delta w = U\Delta\tilde{w} \tag{13}$$

Eqs. 10 through 13 are repeated for every minibatch while DMD is only calculated at every epoch. Note that KGA only accelerates along the selected directions specified by the stable Koopman modes. If there exists no stable modes, then no acceleration is performed.

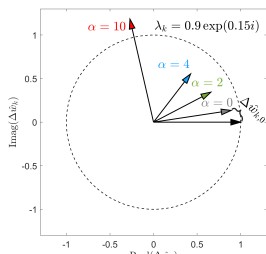

Fig. 3 illustrates the update in the weight of the $k^{\text{th}}$ eigenvector. For a given stable eigenmode, $\hat{w}_k$, the update follows $\Delta\hat{w}_k = (1 + \alpha\lambda_k)\Delta\hat{w}_{k,0}$, where $\Delta\hat{w}_{k,0}$ is the unaccelerated change in the magnitude of the $k^{\text{th}}$ eigenmode when $\alpha = 0$. Intuitively, this algorithm accelerates the decay of Koopman eigenmodes that would otherwise slowly decay to zero. However, as illustrated in Fig. 3, $\alpha$ must be chosen carefully in order to avoid overshooting and amplifying the eigenmodes. The effect of $\alpha$ is further evaluated in Sec. 4.1.

Figure 3: Acceleration of a hypothetical slowly decaying eigenmode $\hat{w}_k$ with associated eigenvalue $\lambda_k = 0.9e^{0.15i}$. Here, $\hat{w}_k = 1$ before the update. When $\alpha = 0$ there is very little change from the update. Moderate values of $\alpha$ hasten decay towards the origin. $\alpha = 10$ overshoots and amplifies this mode.

The pseudocode of the KGA strategy is shown in Algorithm 1, located in Section A.1 of the appendix.

### 3.3 KOOPMAN-BASED WEIGHT-LEVEL MASKING (KWM)

Despite the critical insights brought by analyzing the training dynamics, performing DMD on the weight matrix, $W_i$, can be computationally expensive, especially when the network size is large. Strategies like reduced SVD (Eq. 7) and Efficient-KGA (Eq. 8) alleviate some, but the computation burden is still substantial. The second contribution of this paper is a masking strategy, where based on the contribution of steady modes in reconstruction and in-depth understanding of the Koopman decomposition, we can select only a subset of weights to undergo the gradient acceleration process, resulting in substantial computational savings.

According to Koopman decomposition in Eq. 2, the weight vector at the $i^{\text{th}}$ epoch can be further split into two components according to the type of Koopman modes they use in reconstruction,

$$w_i = \sum_{k=1}^{r} \lambda_k^i c_k(w_0)\phi_k = \sum_{k=1}^{l} \lambda_k^i c_k(w_0)\phi_k + \sum_{k=l+1}^{r} \lambda_k^i c_k(w_0)\phi_k \tag{14}$$

where $r$ is the reduced number of Koopman modes and $l$ is the amount of selected neutral modes. The first component in Eq. 14 corresponds to the weight reconstruction contributed from the "neutral modes" and the second represents those from the "stable and unstable modes". The eigenvalues are sorted by their distance to the unit circle. According to (Redman et al., 2022), the updates on $w_i$ are predominantly dictated by the second component. It is then logical to deduce that if the first component overwhelmingly characterizes a specific weight, then the updates contributed from the second component can be considered inconsequential with no need to go through the KGA procedure. To avoid accidentally masking out weights that temporarily undergo no updates, we adopt a window size of $s$ and only drop weights when there are no major updates for the past $s$ consecutive epochs. This is formulated as follows, where at epoch $i$, the mask that determines if a specific weight $j$ should be masked out is defined by:

$$\text{mask}_j = \begin{cases} 0 & \forall i' \in [i-s+1, i], \ S_{i,j} < \epsilon \\ 1 & \text{otherwise} \end{cases} \quad \text{and } S_{i,j} = \left| \sum_{k=l+1}^{r} \lambda_k^i c_k(w_0)\phi_{k,j} \right| \tag{15}$$

where $\epsilon$ denotes a small positive value to evaluate if the updates contributed from the neutral and unstable modes can be neglected or not.

The integration of the KWM and the KGA strategies is outlined in Algorithm 2 (Sec. A.1 of the Appendix). An element-wise multiplication is conducted between $w_i$ and $mask$ and only those weights remaining will go through the KGA process. Note that the DMD computation is required only once in each epoch, for both KGA and KWM. Once computed, the resulting mask is retained

and applied subsequently to the weights for the upcoming epoch. As training progresses, the number of parameters/weights left for KGA drastically reduces.

# 4 EXPERIMENTS AND RESULTS

We evaluate the proposed Koopman-based training accelerations using four network structures, including a custom-designed multi-layer perceptron (MLP) network of 1K parameters, a custom-designed fully-convolutional network (FCN) of 4M parameters, AlexNet (Krizhevsky et al., 2009) with 60M parameters, and ResNet-50 (He et al., 2016) with 75M parameters. Refer to Sec. A.2 for detailed descriptions of the custom-designed networks.

We perform two sets of experiments. The first set conducts a thorough evaluation of variants of KGA and its performance as compared to prediction-based Koopman acceleration. The second set evaluates the effectiveness of KWM and its impact on KGA.

## 4.1 EVALUATION OF KGA

**KGA-Iter vs. KGA-Epoch.** KGA recalculates $\tilde{A}$ using Eqs. 7, 9 at regular intervals to better capture recent training dynamics. We experiment with using an optimizer-step interval, called KGA-Iter, and an epoch-level interval, called KGA-Epoch. While more frequent recalculations of the $\tilde{A}$ matrix captures the dynamics more effectively, this comes at the cost of increasing the size of $W_i$.

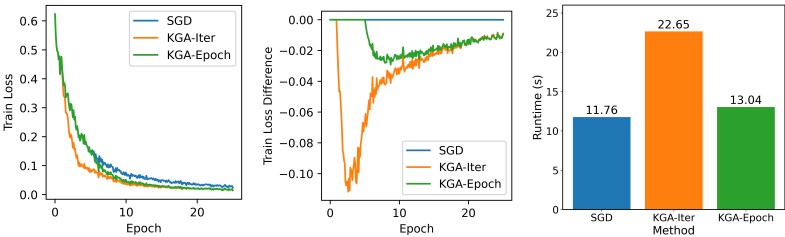

Figure 4: Performance comparison of KGA-Iter and KGA-Epoch with SGD on the small MLP network using the function approximation task of $z = x^2 + y^2$, $x, y \in [-6, 6]$ .

Performance of KGA-Iter and KGA-Epoch is compared with SGD in Fig. 4. We use mean squared error as the objective function and train for 25 epochs. KGA-Iter shows the best improvement in loss, but incurs a significant runtime increase (almost doubled as compared to that of SGD). In the subsequent evaluations, unless specifically mentioned, KGA refers to the KGA-Epoch realization for the tradeoff between runtime and better analysis of system dynamics.

**KGA vs. Efficient-KGA.** Efficient-KGA (Eq. 8) functions similarly to KGA in that it recalculates $\tilde{A}$ only on every epoch, but instead of maintaining a high-dimensional snapshot matrix $W_i$ for the entire training process, we only keep a reduced-order approximation. This reduces the complexity of spacial structures discovered during the training process, but still allows for the discovery of new patterns and trends that can appear during training.

We show the effect of KGA and Efficient-KGA on the AlexNet model in Fig. 5 and comprehensive results on other models are shown in Fig. 10 (Sec. A.4) with all models converging to a superior optima with higher validation accuracy. We also compare the GPU memory usage in Fig. 5. The reduced form of $W_i$ is calculated at epoch 5 in Efficient-KGA. KGA, on the other hand, must continue to accumulate memory to sustain $W_i$.

**KGA vs. Prediction-based Koopman Acceleration.** We utilize an FCN model on the CiFAR10 dataset to compare the performance between KGA and state-of-the-art prediction-based Koopman acceleration (Dogra & Redman, 2020; Tano et al., 2020). In Sec. 1, we discussed the difference between KGA and the prediction-based method and show here that although these methods perform well on smaller networks as the reference show, they would fail for relatively larger networks (even an FCN). The comparison is shown in Fig. 6(a). For this experiment, we use the Adam optimizer for the first 10 epochs to obtain a DMD that appropriately captures the training dynamics, then per-

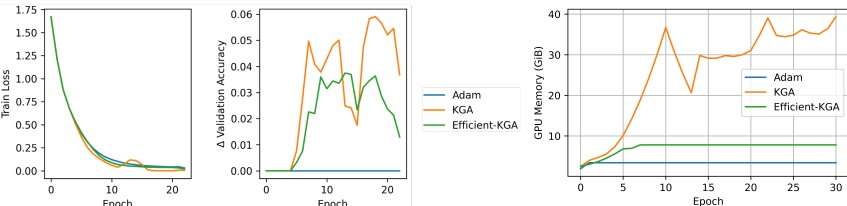

Figure 5: Performance evaluation of KGA and Efficient-KGA compared to Adam when training the AlexNet model on CiFAR-10. Experiment performed on an Nvidia RTX A6000.

form prediction until the validation loss starts to increase. The prediction-based approach (Dogra & Redman, 2020) only has a limited horizon, leading to a sharp decrease in accuracy. The "bootstrap" approach allows the prediction-based approach (Tano et al., 2020) to fall back to Adam when the validation error starts to increase. However, the algorithm still fails to converge to an optimum comparable to that of Adam, and begins to exhibit oscillatory behavior. The proposed KGA, on the other hand, presents consistent performance improvement over Adam.

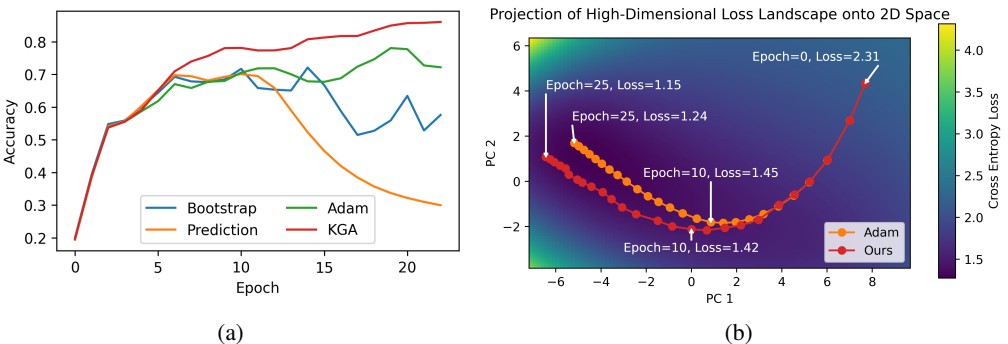

Figure 6: (a) Comparison of the validation accuracy among Adam as baseline (in green), prediction-based acceleration (Dogra & Redman, 2020) (in orange), that with fall-back option ( bootstrap training (in blue) (Tano et al., 2020), and KGA (in red). (b) Comparison of loss trajectory between Adam and KGA. Here, PCA is used to reduce the high-dimensional weight space of AlexNet down to two dimensions where we modify the weights for each point in the figure and evaluate to obtain the point's corresponding validation loss, providing a glimpse into the model's "loss landscape".

Fig. 6(b) provides another view of the superior performance of KGA as compared to Adam, where we accelerate an AlexNet model (Krizhevsky et al., 2012) on the CiFAR10 dataset and show the trajectory through the loss landscape. Note that as KGA accelerates the weight change along stable eigenvectors, its trajectory deviates from that of the Adam optimizer and approaching a better minimum.

**Effect of Hyperpameters.** We demonstrate the effect of $\alpha$ from the update rule in Eq. 12 by sweeping $\alpha$ from $[0 - 20]$ and measuring the difference in the accuracy after 2 epochs, shown in Fig. 9 of Sec. A.3. This experiment is run on the CiFAR10 dataset using the FCN model for 9 epochs.

We benchmark this method against the simple case of increasing the learning rate by running a grid search hyperparameter optimization on the learning rate of the FCN network, optimizing the validation accuracy after 30 epochs using the CiFAR10 dataset. We find the optimal learning rate to be $5.00 \times 10^{-4}$, and compare our method with this as the baseline as well as with learning rates of $1 \times 10^{-5}$, $5 \times 10^{-5}$, $1 \times 10^{-4}$, and $1 \times 10^{-3}$. Results of this experiment are shown in Fig. 8, found in Sec. A.3 of the Appendix. By changing the learning rate, specifically multiplying it by two, all components of the weight update are accelerated. Our method differs in that it only accelerates the stable, slow moving components of the weights.

## 4.2 EVALUATION OF KWM

The second set of experiments demonstrate the efficacy of KWM in retaining the benefits of KGA while significantly reducing computation. In addition, we introduce two straightforward but effective masking strategies as baseline, 1) Norm Masking that masks individual weights with norm values below a specified threshold and 2) Delta Masking that masks weights exhibiting minimal changes over the past $N$ epochs. We integrate these three masking strategies with the proposed KGA method and compare the performance by training the FCN network on CiFAR10. The results are shown in Fig. 7. More results on AlexNet and ResNet-50 are shown in Fig. 12 of Sec. A.4.1. Acceleration is applied after the 5th epoch. The presented results include the training loss, the loss difference (using the naive optimizer without KGA as a baseline), the masking ratio, and the additional computation time required for DMD. Across all models, we observe that as a larger fraction of weights is masked, computation time is correspondingly reduced. As training progresses, the computational savings from masking become increasingly significant. Notably, the optimal balance between execution time and performance is contingent on the specific architecture, task, and hyperparameter selection. In Sec. A.5, we demonstrate that as individual weights are close to convergence, KWM provides a general explanation to both delta masking and norm masking.

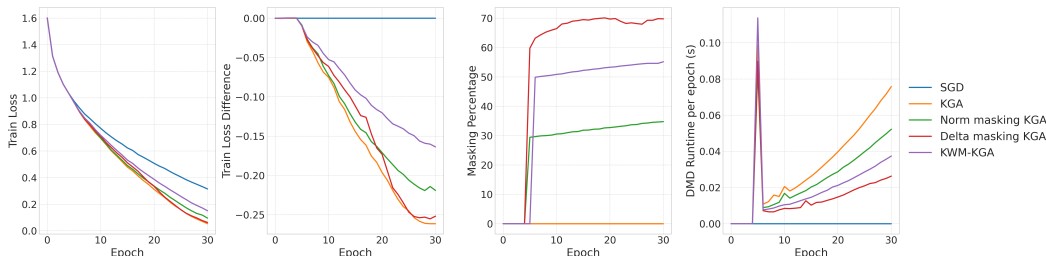

Figure 7: Performance of Masked KGA on FCN using Norm Mask, Delta Mask, and KWM.

Table 1: Runtime comparison after training AlexNet on CiFAR10 for 30 epochs. Experiments performed on an Nvidia RTX A6000. Although three masking methods reduce the GPU memory usage and DMD computation time by reducing the matrix size for DMD, they don't always offer runtime benefits due to the time needed for mask processing. When more training epochs are required, and the matrix size for DMD increases, masking methods can become advantageous in runtime.

| | Runtime (s) | |
| Method | SGD as Baseline | Adam as Baseline |
|---|---|---|
| Baseline | 338.15 | 343.60 |
| KGA | 362.91 | 390.81 |
| Norm Masking KGA | 394.33 | 414.00 |
| Delta Masking KGA | 362.64 | 399.14 |
| KWM-KGA | 403.28 | 443.06 |

## 5 CONCLUSION

In this paper, we proposed a novel set of training strategies that incorporates Koopman operator theory to further understand the intricacies of the dynamics in deep neural network training. Building on the inherent dynamics of the model training process, we developed two acceleration techniques: Koopman-based gradient acceleration (KGA) and Koopman-based weight-level masking (KWM). The KGA technique exploits the dynamics of the training process itself to identify and accelerate slow-moving structures. This acceleration guides the gradient to the descent along these stable Koopman eigenmodes. The KWM technique, on the other hand, reduces the computation overhead of KGA by selecting only a subset of weights contributed from stable modes. The proposed method not only achieves better performance but also converges faster than traditional training methodologies. Moreover, our approach is designed to be easily extended to most models and training processes, regardless of data distribution or batch dependence.

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

APPENDIX TABLE OF CONTENTS

# A   APPENDIX

## A.1   ALGORITHM DESCRIPTIONS

---
**Algorithm 1** The KGA Strategy
---
**for all** epochs $i$ **do**
   **if** $i \leq r$ **then**
      Do not accelerate
   **end if**
   Calculate DMD for $W$ (*Eqs. 7,9*)
   **for all** minibatch steps **do**
      $\Delta w \leftarrow (w_{i+1} - w_i)$ (*Optimizer step*)
      $\Delta \tilde{w} \leftarrow U^T \Delta w$
      $\Delta \hat{w} \leftarrow L_{\text{stable}} \Delta \tilde{w}$
      $\Delta \tilde{w} \leftarrow \Delta \tilde{w} + \alpha \left( Q_{\text{stable}} \Lambda_{\text{stable}} \Delta \hat{w} \right)$
      $\Delta w \leftarrow U \Delta \tilde{w}$
   **end for**
**end for**
---

---
**Algorithm 2** The KWM Strategy
---
**for all** epochs $i$ **do**
   **if** $i \leq r$ **then**
      Do not accelerate
   **end if**
   Use the extant weight mask to obtain $W_m$
   Calculate DMD for $W_m$ (*Eqs.7, 9*)
   Perform KGA on $W_m$ (Alg. 1)
   Decompose $W_m$ (*Eq.14*)
   Evaluate the score for individual weights in $W_m$ and update to a new mask. (*Eq. 15*)
**end for**
---

## A.2   IMPLEMENTATION DETAILS

**Customized Network Structure.**   The fully-convolutional network is designed for classification and comprises five convolutional layers, successively increasing in the number of filters from 64 to 512, each with $3 \times 3$ kernels. The final layer is a $1 \times 1$ convolution that maps the 512-dimensional feature map to the correct number of classes. This network also utilizes batch normalization and global average pooling. The MLP network, on the other hand, is designed for regression, and comprises 3 layers with 32 neurons in the hidden layer.

**Vectorized DMD.** To perform SVD for the large $W \in R^{m \times t}$ matrix where $m$ can be on the order of millions and $t$ can be tens or hundreds, we offload these computations to the GPU and utilize the cuSOLVER routines from the CUDA toolkit.

## A.3 Effect of Hyperparameters

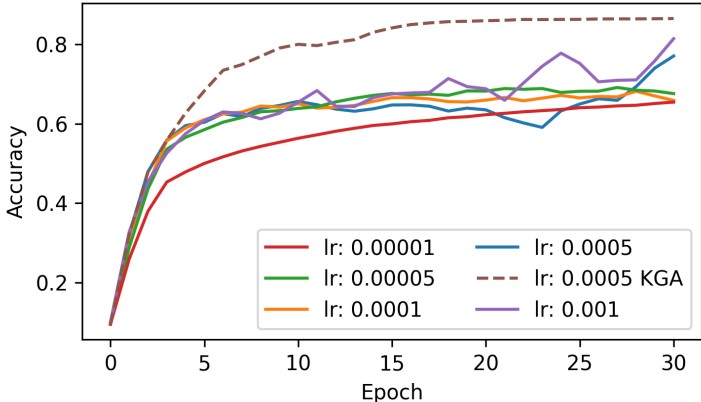

Figure 8: Validation accuracy of CiFAR10 using the FCN network trained at different learning rates vs. our method with the learning rate found with HPO as the baseline.

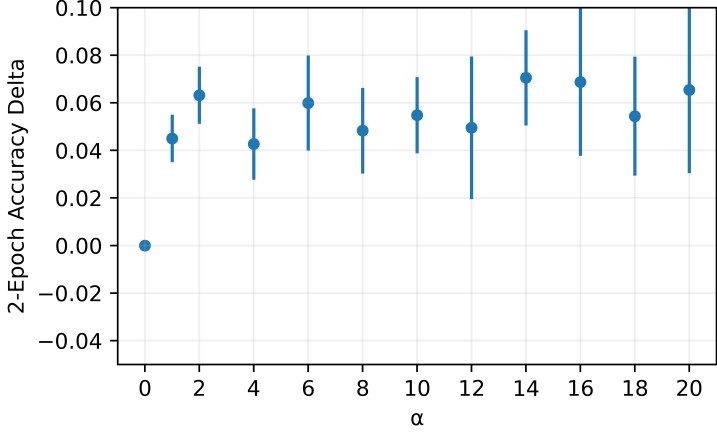

Figure 9: A scatterplot with values of $\alpha$ on the x-axis and performance, expressed as the difference in loss compared to the Adam optimizer, on the y-axis. Vertical bars represent the standard deviation over 3 runs with different initialization. This experiment is performed on the FCN model and the CiFAR10 dataset.

## A.4 ADDITIONAL NETWORK STRUCTURES

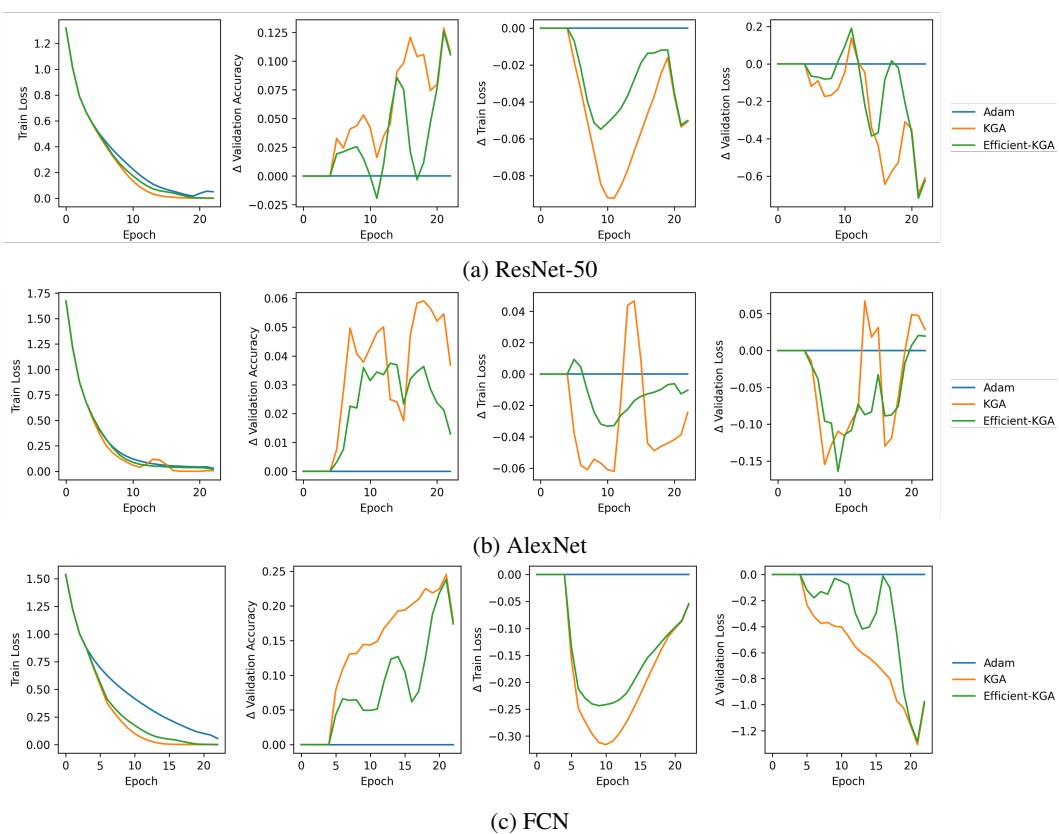

Figure 10: Performance of KGA-Epoch and Efficient-KGA compared to Adam on ResNet-50, AlexNet, and FCN.

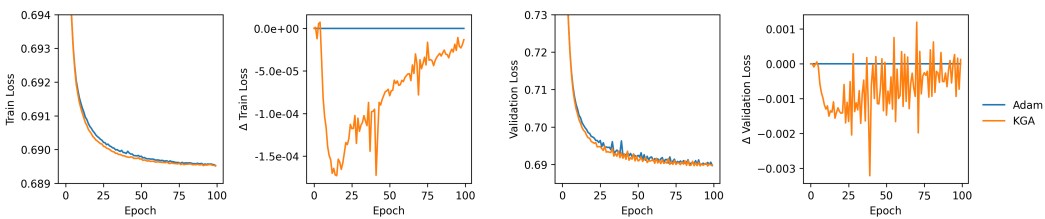

Figure 11: Performance of KGA-Epoch compared to Adam on a CNN-based Variational Autoencoder Kingma & Welling (2013).

### A.4.1 COMPUTATION SAVING ON OTHER MODELS

The integrated results of KGA and KWM on AlexNet and ResNet are provided in Fig. 12 These two models converge at earlier epochs. The KGA preserves the advantages in loss difference before convergence. Fig. 12(c) using Momentum as the baseline optimizer to further demonstrate the compatibility and enhancement effect of KGA in conjunction with various base optimizers.

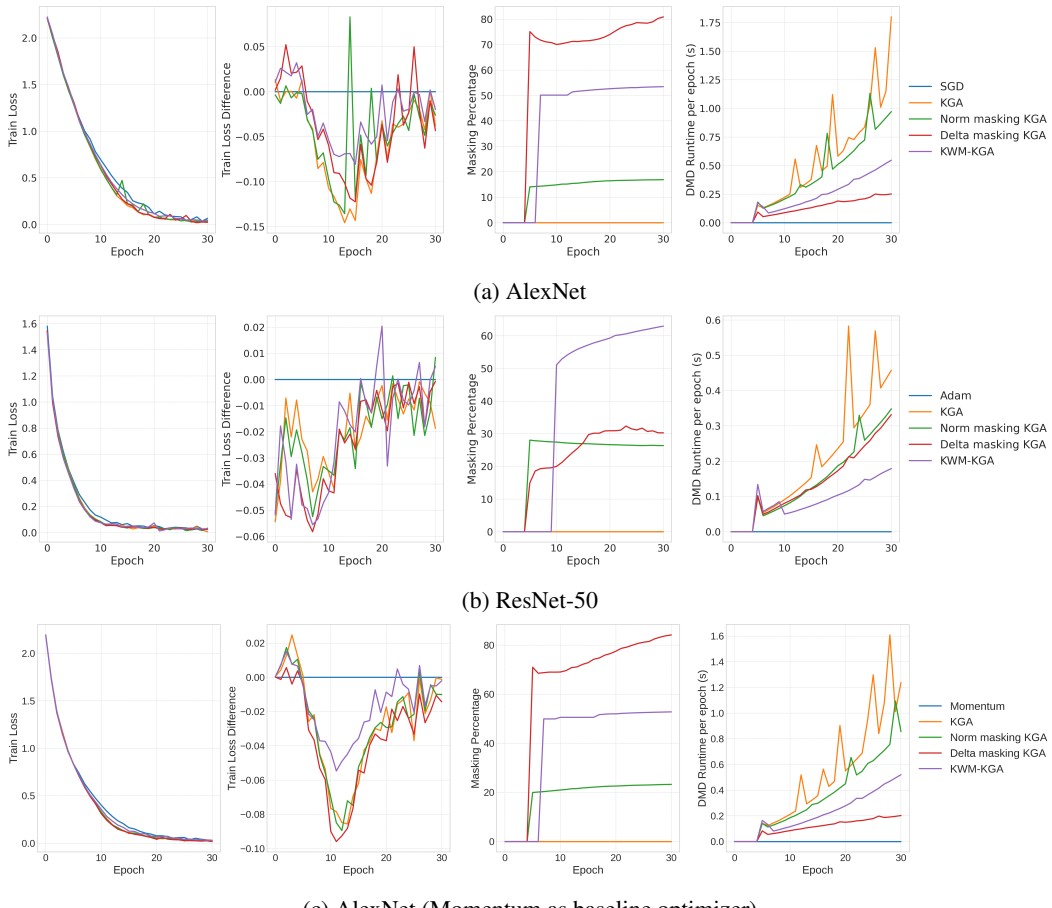

(a) AlexNet

(b) ResNet-50

(c) AlexNet (Momentum as baseline optimizer)

Figure 12: Performance of Norm Mask, Delta Mask, and Koopman-based Mask on AlexNet and ResNet-50 using CiFAR10 dataset.

## A.5 EQUIVALENCE OF THE THREE MASKING STRATEGIES

### I. KWM and Delta Masking

- **a) Delta Masking Application:** Fig. 13 (a) depicts a weight masked via delta masking. The red dashed line indicates the moment the weight is masked. This occurs when the weight's variation remains below a specified threshold for the past 5 epochs.

- **b) KWM Application:** Fig. 13 (b) illustrates a weight masked by KWM. The blue line represents the reconstruction contributed by the neutral mode, corresponding to the first term in Eq. 14. As observed, when a weight approaches the blue line and maintains this proximity for a window size of 5, it is subject to masking.

- **Comparison:** Delta masking evaluates a weight against its historical values within a given time window. Conversely, in KWM, a weight is contrasted with the reconstruction derived from neutral modes over a similar duration. As the weight is close to convergence, the neutral mode reconstruction approximates a straight line. Consequently, weights masked by KWM naturally fulfill delta masking criteria. This behavior demonstrates how KWM encompasses Delta Masking.

### II. KWM and Norm Masking

- **c) Norm Masking Application:** Fig. 13 (c) shows a weight masked by norm masking. When the weight value diminishes sufficiently, it undergoes masking.

- **d) KWM Application:** Fig. 13 (d) showcases a weight masked using KWM. It underscores that if a weight is on the brink of converging to a minimal value, its reconstruction from the neutral mode will also be small. Such behavior invariably meets the norm masking criteria.

- **Comparison:** This similarity in behavior strongly suggests how KWM can effectively interpret Norm Masking.

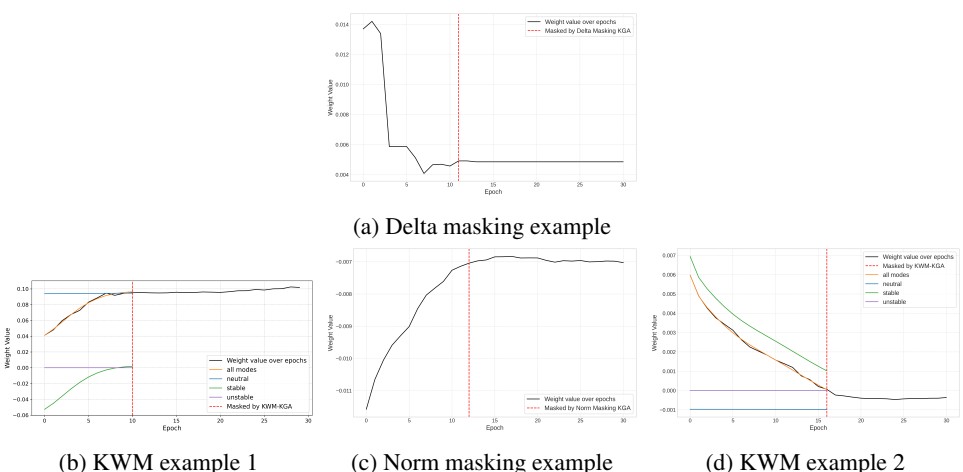

(a) Delta masking example

(b) KWM example 1      (c) Norm masking example      (d) KWM example 2

Figure 13: Examples of Delta masking, Norm masking, and KWM. The weights from AlexNet using SGD. The vertical line indicates where the weight was masked from acceleration.

### A.6 ASSUMPTIONS OF EFFICIENT-KGA

Efficient-KGA assumes that the optimization dynamics are smooth and that the singular vectors will continue to capture a significant amount of variance after U is calculated and stored. While this may not be true for many dynamical systems, we show that for neural network training dynamics, singular vectors from early epochs still capture a significant amount of later epochs' variance. We measure the captured variance as $var(UU^T w_i)$ where $w_i$ is the weight vector from the $i$'th epoch. The proportion is then $\frac{var(UU^T w_i)}{var w_i}$.

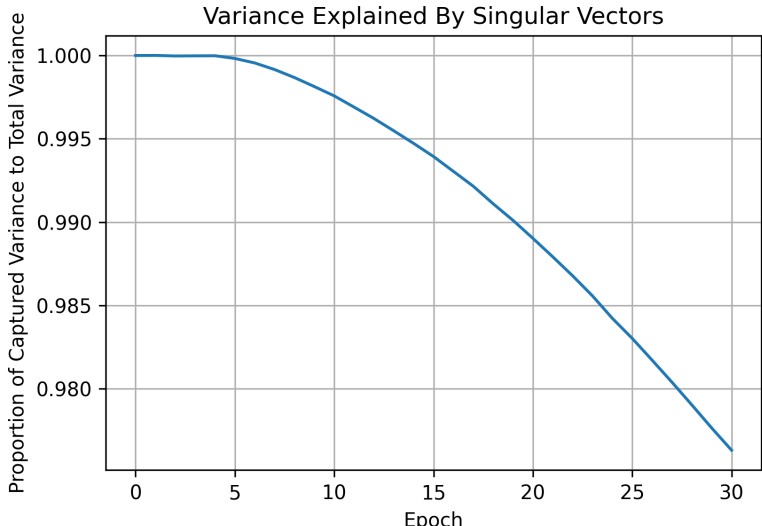

Figure 14: Proportion of variance explained over time by singular vectors from epoch 5. Each point on the graph represents the proportion of the total variance in the data at a specific time point that is explained by these reference singular vectors. A high value indicates that the reference singular vectors effectively capture the essential features and variability of the data at that time point. Data from KGA training AlexNet on the CiFAR10 dataset.

