# OpenReview forum: "Dynamic Training Guided by Training Dynamics"
_ICLR.cc/2024/Conference — Submitted to ICLR 2024_

### Official Review · Reviewer_APZy · 2023-10-31

**Soundness:** 3 good
**Presentation:** 3 good
**Contribution:** 3 good
**Rating:** 5
**Confidence:** 2

**Summary:**

This paper proposed an adaptive weight updating method for gradient-based training of neural networks. The idea is to model the trajectory of network weights during training and then extrapolate to future steps/epochs. The trajectory is modelled using Dynamic Mode Decomposition (DMD). This can be reviewed as a look-ahead trick to speed up convergence. The authors showed that the additional prediction of weight changes lead to faster covergence in terms of the number of epochs in some cases.

**Strengths:**

- The proposed algorithms seem to be technically sound.

**Weaknesses:**

- The methods and experiments do not seem to completely described. With no code provided, I had to guess and fill in potential details myself.
- It is unclear whether the proposed methods indeed make consistent improvement compared with SOTA optimisers. The empirical comparisons were not carefully designed, with arbitrary and inconsistent selection of baselines. For example, Adam was compared, but it is also used as a fall-back when KGA (the proposed method) does not seem to help. In addition, the KGA seems to have a significant memory footprint and requires additional running time; there should be fair comparison with the SOTA optimisers, under the same budget of computing resources.
- Section 3.3 seems to be there just to add more technical content. It is not clear to me whether it actually work significantly better than simpler baselines such as Delta masking or Norm masking.

**Questions:**

- I do not understand Figure 3. Why does alpha=10 mean the vector is outside the unit circle? How does it related to Figure 12?
- In equation 14, how are lamdas indexed? What are "steady" modes?

---

> ### Author Response · Authors · 2023-11-23
> **Response to Reviewer APZy**
>
> > **Weakness 1** - Regarding the code of proposed methods.
> >
> Pseudocodes for KGA and KWM have been provided in the Appendix A.1.
>
> > **Weakness 2** - Regarding the tradeoff between cost vs. performance gain and the consistency of experiments:
> >
> We completely agree that the evaluation between cost and performance gain should be the center of experiments and results. During the rebuttal phase, we spent quite some effort to streamline the discussion of experiments and the way the results are presented. Here’s a summary of changes related to the cost vs. performance gain comparison.
> 1. We revised Fig. 4 by moving Table 1 (originally in Appendix A.3) to be a subfigure in Fig. 4, such that the left two subfigures show the performance gain of KGA as compared to SGD (in terms of training loss), and the right subfigure shows the runtime cost of KGA-Iter. Through this comparison, we conclude the better tradeoff achieved using the KGA-Epoch approach.
> 2. We revised Fig. 5 by selecting the most informative results from Figs. 8 and 11 (originally in Appendix A.3) , such that the left two subfigures show the performance gain of Efficient-KGA in terms of validation accuracy and the right subfigure shows the memory cost. Through this comparison, we conclude the better tradeoff achieved using the Efficient-KGA over the KGA-Epoch approach.
> 3. We revised Fig. 6 by moving Fig. 10 (originally in Appendix A.3) as a subfigure to the left. It shows the performance gain of Efficient-KGA over state-of-the-art prediction-based Koopman acceleration approach.
> 4. We conducted an additional experiment during the rebuttal phase to provide a comprehensive runtime comparison for the integrated strategy of masking+KGA. This is added to the main paper as Table 1. This complements the performance gain comparison in Fig. 7.
>
> > **Weakness 3** - Regarding Section 3.3, the comparison of KWM, Delta masking, and Norm masking:
> >
> Sec. 3.3 is there to describe a masking strategy based on the analysis of Koopman-based training dynamics. It is necessary to further reduce the cost of having to perform SVD on a large matrix. The KWM approach can be regarded as a generalized version of some existing ad-hoc masking approaches, like delta masking and norm masking. This is demonstrated in the Appendix A.5.
>
> > **Question 1** - I do not understand Figure 3. Why does alpha=10 mean the vector is outside the unit circle? How does it related to Figure 12?
> >
> In the theoretical example shown in Figure 3, moderate values of $\alpha$ result in acceleration, the change in angle is a result of the underlying optimization algorithm (SGD/Adam). Choosing a large value of $\alpha$ will result in accelerating the eigenvalue/eigenmode pair to become unstable, resulting in a significant drop in performance. In Fig. 9 of the Appendix, we show empirical results after training with various values of $\alpha$ using an FCN model on Cifar 10. The figure shows that using higher values of $\alpha$ yields larger standard deviations in accuracy, indicating that the performance increase is not consistent with large values.
>
> > **Question 2** - In equation 14, how are lamdas indexed? What are "steady" modes?
> >
> To clarify and avoid any potential misunderstandings, we have decided to revise the terminology in our manuscript. The term 'steady modes' has been replaced with 'neutral mode' to better convey our intended meaning in the context. Regarding the indexing of lamdas, they are sorted based on their distance to the unit circle.
> We appreciate your attention to detail and have revised the paper to correct any ambiguities and typographical errors. Your feedback has been instrumental in enhancing the clarity and accuracy of our paper.

---

### Official Review · Reviewer_ptUk · 2023-10-31

**Soundness:** 3 good
**Presentation:** 1 poor
**Contribution:** 1 poor
**Rating:** 3
**Confidence:** 3

**Summary:**

This work introduces a method aimed at accelerating the convergence of neural network training. Specifically, the proposed approach applies Singular Value Decomposition (SVD) to decompose a set of weights at a given timestep, denoted as W_i. Through this process, the method identifies more important and less important basis components within the weights. Subsequently, the authors project the weights of the next timestep, W_{i+1}, onto these basis components. Finally, they emphasize the major axis (important basis) when calculating the update term (W_{i+1}-W_{i}). This method effectively adds weighting to the important basis components, contributing to faster convergence during training.
However, I have some concerns as follows:

1)	It appears that the primary focus of this work is identifying important directions within the current weights (W_i) and emphasizing updates along these directions to facilitate faster convergence during training. However, the background and related works seem to be rooted in dynamic systems knowledge. I think the introduction and related works are not fit to the proposed work.

2)	The method proposed in this work involves storing the matrix U for efficient computation. However, this approach makes an assumption that the important directions within the weights exhibit relatively fewer changes over a certain number of iterations or epochs. This assumption is essential for the method's effectiveness, but it should ideally be substantiated through either theoretical or empirical evidence to validate this point and applicability in practice.

3)	Additionally, it's worth noting that the KWM method also reuses a mask for an entire epoch to select candidates, which entails a similar assumption about the stability of important directions within that epoch.

4)	Several prior works have applied SVD or Eigen Decomposition to identify important weight components, often for purposes such as pruning or interpretation. It appears that the proposed method shares similarities with training a pruned networks (KWM) with higher learning rates (KGA), or even lifelong training from a small network. To provide a more comprehensive understanding of the proposed method, it would be valuable to conduct a comparison with these related approaches in terms of their underlying ideas and concepts, even if the focus is not solely on performance comparison. This would help clarify the method's novelty and its place within the existing body of research.

5)	While the paper compares the proposed method with Adam, SGD, and some variants of the proposed approach, it's important to note that there are various other methods and algorithms designed for faster trainin. To provide a more comprehensive assessment of the benefits of the proposed method, additional experiments and comparisons with a broader range of optimization algorithms and training techniques would be valuable.

6)	Given the similarity between the proposed method and using a higher learning rate for specific weight components, it is important to consider the learning rate issue in the comparison experiments. The experiments provided in the paper, where the same learning rate was used for both the proposed method and the naive optimizers, may not provide a fair comparison.

7)	From my understanding, this work utilizes SVD for the weights of a single timestep, rather than considering multiple timesteps. This approach appears to capture the state of the network at a single previous step, rather than modeling training dynamics that involve interactions across multiple timesteps. Clarification or further exploration of the method's relationship to training dynamics may be beneficial to provide a more comprehensive understanding of its operation and potential limitations.

8)	In minor, the citation style is different from other papers.

**Strengths:**

See above

**Weaknesses:**

See above

**Questions:**

See above

---

> ### Author Response · Authors · 2023-11-23
> **Response to Reviewer ptUk (part 1/2)**
>
> Thank you for your insightful comments.
>
> > **Comment 1** - Regarding the background and related works:
> >
> The reviewer is correct in that the primary focus of the paper is to identify the important directions (i.e., the slowly decaying modes) along which we could selectively accelerate the weight update. The method we used is based on the fundamental idea of treating the training process as a dynamical system. We use Koopman operator theory (a data-driven dynamical system analysis approach) to extract training dynamics and use it to find the acceleration direction. As such, the introduction and especially the related work focuses on the Koopman analysis approach. Eqs. 1-6 in Sec. 2 are essential to understand the Eqs. 7-15 in the Methods section.
>
> > **Comment 2** - The method proposed in this work involves storing the matrix U for efficient computation. However, this approach makes an assumption that the important directions within the weights exhibit relatively fewer changes over a certain number of iterations or epochs. This assumption is essential for the method's effectiveness, but it should ideally be substantiated through either theoretical or empirical evidence to validate this point and applicability in practice.
> >
> For the case of Efficient-KGA, it is true that we make the assumption that the singular vectors of $W_i$ after $n$ epochs sufficiently capture the variance for the rest of training. Additional experiment has been conducted to empirically support this claim. Fig. 14 in Appendix A.6 shows that the singular vectors calculated in epoch 5 continue to explain 98% of the variance.
>
> > **Comment 3** - Additionally, it's worth noting that the KWM method also reuses a mask for an entire epoch to select candidates, which entails a similar assumption about the stability of important directions within that epoch.
> >
> Yes, the reviewer is right that KWM uses the same mask for each epoch. KWM is designed to not involve additional computation. Therefore, it always coordinates with the same time resolution as KGA, ensuring efficiency and consistency in our approach.
>
> > **Comment 4** - Several prior works have applied SVD or Eigen Decomposition to identify important weight components, often for purposes such as pruning or interpretation. It appears that the proposed method shares similarities with training a pruned network (KWM) with higher learning rates (KGA), or even lifelong training from a small network. To provide a more comprehensive understanding of the proposed method, it would be valuable to conduct a comparison with these related approaches in terms of their underlying ideas and concepts, even if the focus is not solely on performance comparison. This would help clarify the method's novelty and its place within the existing body of research.
> >
> **Similarities with existing methods:**
> 1. SVD-Based Analysis: KWM utilizes SVD to monitor the evolution of individual weights during training, hence akin to some interpretation and pruning methods.
> 2. Selective Weight Operations: Similar to methods that evaluate the importance of individual weights, KWM identifies a subset of weights for differential treatment.
>
> **Unique Features of KWM:**
> 1. Online Masking Strategy: Unlike retrospective methods, KWM’s online masking strategy is driven by the need to accelerate training in real-time.
> 2. Selective Acceleration without Pruning: Weights within the mask benefit from acceleration, while those outside are trained using the default optimizer. This means they are neither frozen nor pruned, making KWM a more conservative approach that aims to preserve performance.
> 3. Interpretative Potential: KWM can potentially be used to assess the convergence status of individual weights during training, which can serve both interpretative and training-guidance purposes.
> 4. Adaptation of Koopman Analysis: While KWM uses Koopman analysis to extract unstable modes, it has been modified to suit the online strategy, distinguishing it from traditional approaches.
>
> We hope this explanation clarifies the distinctive aspects of KWM and its contribution.

---

> ### Author Response · Authors · 2023-11-23
> **Response to Reviewer ptUk (part 2/2)**
>
> > **Comment 5** - While the paper compares the proposed method with Adam, SGD, and some variants of the proposed approach, it's important to note that there are various other methods and algorithms designed for faster training. To provide a more comprehensive assessment of the benefits of the proposed method, additional experiments and comparisons with a broader range of optimization algorithms and training techniques would be valuable.
> >
> In general, the proposed KWM-KGA is designed to function ***on top of*** any underlying network optimization paradigm. Our original experiments utilize SGD and Adam as baseline and apply KGA on top of these methods. We have added an experiment with Momentum as the baseline (added in Figure 12(c) in Appendix A.4.1) to further demonstrate the compatibility and enhancement effect of KGA in conjunction with various base optimizers. While we understand that a more extensive range of experiments and comparisons might provide a more comprehensive assessment, given that our method aims to enhance rather than replace existing optimization techniques, we hope the current experimental setup sufficiently supports our assertions.
>
> > **Comment 6** - Given the similarity between the proposed method and using a higher learning rate for specific weight components, it is important to consider the learning rate issue in the comparison experiments. The experiments provided in the paper, where the same learning rate was used for both the proposed method and the naive optimizers, may not provide a fair comparison.
> >
> We agree with the reviewer that the effect of learning rate is an important factor to evaluate. We actually included this experiment in the original paper. However, due to the page limit, we had to place the results in the Appendix (Fig. 8 in Appendix A.3) and only leave the description and findings from the experiments in the main paper. For each (model, dataset, optimizer) combination, we performed a hyperparameter optimization (HPO) to determine the optimal learning rate for that combination. The optimal learning rate was used in all experiments, with KGA and the baseline using the same learning rate. For example, Fig. 8 shows the optimal learning rate with HPO training the FCN on CiFAR10 is  $5 \times 10^{-4}$. We compare KGA with this learning rate as the baseline as well as with learning rates of $1 \times 10^{-5}$ , $5 \times 10^{-5}$, $1 \times 10^{-4}$, and $1 \times 10^{-3}$. Fig. 8 shows that KGA with the optimal learning rate performs superior than simply using different learning rates. KGA differs in that it only accelerates the stable, slow-moving components of the weights.
>
> > **Comment 7** - From my understanding, this work utilizes SVD for the weights of a single timestep, rather than considering multiple timesteps. This approach appears to capture the state of the network at a single previous step, rather than modeling training dynamics that involve interactions across multiple timesteps. Clarification or further exploration of the method's relationship to training dynamics may be beneficial to provide a more comprehensive understanding of its operation and potential limitations.
> >
> This work utilizes SVD to perform a proper orthogonal decomposition on $W_i$. The matrix actually contains weights of the network over the last t epochs. Hence, the interactions across multiple timesteps (or epochs) are indeed captured in the analysis process.
>
> > **Comment 8** - In minor, the citation style is different from other papers.
> >
> Thanks for pointing it out. We were able to fix the problem in the revised paper.

---

### Official Review · Reviewer_xUqE · 2023-10-31

**Soundness:** 2 fair
**Presentation:** 2 fair
**Contribution:** 2 fair
**Rating:** 3
**Confidence:** 5

**Summary:**

The work builds upon and investigates prior Koopman operator theory (KOT) based approaches to analysing and boosting ML optimization, taking advantage of the linearized perspective offered by KOT in conjunction with the operator compression methods like DMD to obtain adequate low rank estimates. They propose an algorithm for turning the noted connections between ML optimization and KOT in prior works into practical application by creating a gradient boosting strategy that is further augmented by a masking strategy to lower the burden of the needed Koopman computations. Numerical results are presented to support claimed results.

**Strengths:**

A. This work builds well on prior touchstone papers - showcasing why their insight was useful and applicable, while also showcasing the limits of those early insights. In particular I am pleased to see the comparison to Dogra and Redman 2020, which was the second (see below) systematic proposal and exploration of these connections, yet severly limited in actual practical impact due to the shorter time period of effectiveness of the estimated Koopman operators (their method has a tendency to veer off-course significantly earlier than this work's as shown by the authors).


B. The proposed method consistently outperforms on accuracy for the chosen tasks (for caveat, see below), and for reasons that are borne out by the theoretical development well. It is generic enough to scale to many problems as well (for caveats, see below).

C. The scale of the numerical experiments compares very well compared to the older papers (75M parameters vs order 10k parameters in the older works).

**Weaknesses:**

A. In my opinion, by far the most significant weakness is how the cost vs performance gains comparison is handled. For ex, consider Fig. 7d, which is missing the comparison to the costs incurred in SGD (the blue curve is invisible). The proposed method outperforms in accuracy, but if the costs are astronomically high, are those gains worth it? Indeed, this should be tabulated and presented as a central focus of the work (whether in the limitations section or the advantages section I don't know, because I couldn't easily see what the overall cost versus performance trade-offs were). My score will change substantially depending on what the true results are (and my deepest apologies are extended to the authors if I have simply missed the comparison somehow in the main body - which is itself a weakness of the work as well, since this comparison should be unmissable). Table 1, A.3 seems to be the relevant comparison but it being in the appendix is a strong oversight. I am happy to be proven wrong (and change my score aptly) based on this point.

B. The literature review is missing two important early works in the modern ML/KOT intersection. To the best of my knowledge, the idea that KOT could be profitably used to study/boost optimization (especially in ML, but in general as well) was first mentioned in [1] and first properly investigated within the realm of ML in [2]:

[1] Section 4.1, F. Dietrich, T. N. Thiem, I. G. Kevrekidis, On the Koopman Operator of Algorithms, arXiv:1907.10807 (2020)

[2] Section 3, A. S. Dogra, Dynamical Systems and Neural Networks, arXiv:2004.11826 (2020)

C. The applications to which this method has been turned to seem not as impressive as the method or the size of NNs to which it is applied. There is no dearth of substantial ML problems that are intensely resource heavy AND non-trivial - surely the true validation would be to showcase that the method scales well to other kinds of problems too (I am happy to be corrected on this point). The authors do an excellent job of handling large parameter models (if I am right in inferring that 75M means 75 million parameters for ResNet-50). However, as interesting as it is to me to see where the method succeeds is where it would fail - which problems/loss functions are too complicated for the method to hold (or is the method practically so generic that for every type of loss function and problem, it succeeds? That would be a bold claim, even though the theory to KOT does support that: the practice of KOT has seldom shown that however due to how costly it is to compute the needed operators). An easy approach would be to compare to the Hamiltonian solvers Koopman boosted by Dogra and Redman 2020, and show that the authors methods blow the prior work out of the water.

D. While not critical, there are presentation issues in the work. For ex, Fig. 3 is poorly presented and formatted (some symbols are almost impossible to read on the circle. In general the font sizes on all figures should be substantially larger).

**Questions:**

Please see weakness B and answer if such an experiment would be possible.

---

> ### Author Response · Authors · 2023-11-23
> **Response to Reviewer xUqE**
>
> Thank you for your insightful comments.
>
> > **Weakness A** - Regarding the tradeoff between cost vs. performance gain:
> >
> We completely agree that the evaluation between cost and performance gain should be the center of experiments and results. We also agree that this evaluation should not be pushed to the Appendix even with a page limit. During the rebuttal phase, we spent quite some effort to streamline the discussion of experiments and the way the results are presented. Here’s a summary of changes related to the cost vs. performance gain comparison.
>
> 1. We revised Fig. 4 by moving Table 1 (originally in Appendix A.3) to be a subfigure in Fig. 4, such that the left two subfigures show the performance gain of KGA as compared to SGD (in terms of training loss), and the right subfigure shows runtime cost of KGA-Iter. Through this comparison, we conclude the better tradeoff achieved using the KGA-Epoch approach.
>
> 2. We revised Fig. 5 by selecting the most informative results from Figs. 8 and 11 (originally in Appendix A.3) , such that the left two subfigures show the performance gain of Efficient-KGA in terms of validation accuracy and the right subfigure shows the memory cost. Through this comparison, we conclude the better tradeoff achieved using the Efficient-KGA over the KGA-Epoch approach.
>
> 3. We revised Fig. 6 by moving Fig. 10 (originally in Appendix A.3) as a subfigure to the left. It shows the performance gain of Efficient-KGA over state-of-the-art prediction-based Koopman acceleration approach.
>
> 4. About Fig. 7d: It shows the “extra” runtime for calculating DMD in each epoch, which is an additional cost. Since SGD does not need to calculate DMD, it does not have a corresponding value in this figure. As such, SGD would be at y=0 and was left out. We have updated Fig. 7 and Fig. 13 to draw the blue curve corresponding to SGD and Adam.
>
> 5. We conducted an additional experiment during the rebuttal phase to provide a comprehensive runtime comparison for the integrated strategy of masking+KGA. This is added to the main paper as Table 1. This complements the performance gain comparison in Fig. 7.
>
> > **Weakness B**  - Regarding the two missing literature:
> >
> Thank you for suggesting these two references! We were worried we might have missed some important literature, especially in the early stage of KOT and acceleration development. We have added both in Sec. 2 the literature review.
>
> > **Weakness C**  - Regarding limitation in application:
> >
> Dogra and Redman (2020) concluded that Koopman training can be applied to a range of feedforward, fully connected NNs with different architectures, objectives, activation functions, and optimization algorithms (including stochastic ones). In this paper, we extended this in several ways: 1) using much larger networks with millions of parameters, 2) using CNN-based networks (e.g., AlexNet) in addition to traditional multi-layer perceptron (e.g., MLP), and 3) using networks with skip-connections (e.g., ResNet). During the rebuttal phase, we have added an additional experiment training a variational autoencoder to demonstrate the efficiency still holds on a network with a different loss function. This is included as Fig. 11 of Sec. A.4. However, we do agree that it would be very interesting to test out more network structures for extreme applications. For example, it is our intention to continue the experiments on ViT network, on self-supervised learning models (e.g., MAE), and probably on diffusion models (with different schedulers).
>
> > **Weakness D** - Regarding Fig. 3:
> >
> We also realized the difficulty in reading and interpreting Fig. 3. We apologize for this inconvenience. We have redrawn Fig. 3 and modified the description. Basically, we wanted to provide a graphical illustration of how the different acceleration parameters ($\alpha$) might affect the acceleration of an eigenmode. In Sec. 4.1, we provided more comprehensive evaluation of the effect of $\alpha$ and the results are shown in Fig. 9 of Appendix A.3.

---

### Official Review · Reviewer_aC1Y · 2023-11-04

**Soundness:** 1 poor
**Presentation:** 2 fair
**Contribution:** 2 fair
**Rating:** 3
**Confidence:** 3

**Summary:**

This paper suggests two modifications to training algorithms based on Koopman mode decompositions of the weights. Termed Koopman Gradient acceleration, this modification is to project the gradient (or equivalently, the changes in the weight during a training iteration) on the stable singular vectors of the matrix approximation of the Koopman operator. They also propose a more computationally efficient version of KGA, called KWM or Koopman weight masking, where, the components that have negligible unstable Koopman components are set to 0. These modifications are compared against vanilla SGD in training some fully connected and convolutional networks, for a few epochs; in numerical experiments, they are shown to produce smaller training and test errors.

**Strengths:**

* The focus on numerical experiments to demonstrate KGA and the speed up obtained using KWM
*  drawing connections between the optimization and dynamics viewpoints.

**Weaknesses:**

1. My main concern is that the methods described are based on heuristics and no theoretical justification is presented. Hence, it is unclear whether this method will actually lead to convergence to an optimization dynamics (e.g. SGD) in a given non-convex optimization problem.

2. The terms stable manifold etc are used incorrectly -- these do not refer to manifolds tangent to the singular vectors of the Koopman operator approximations. The demarcation into "stable", "unstable" and "neutral" Koopman modes is also quite hand-wavy and it is not explained what projecting the weight updates to the basis of some singular vectors of a matrix approximation of the Koopman operator, actually does.

3. First of all, the DMD matrix is only an approximation of the Koopman operator on L^2. It would change with the snapshots used. Projecting on the range of a particular approximation induces a representation error and further the ad hoc masking completely changes the optimization dynamics.

**Questions:**

All the major questions are about the soundness of the method proposed and why it is expected to work.

1. Please check the equations for the Koopman mode decompositions (the DMD part seems correct to me). It should read as $$K g(x) = \sum_{k \in \mathbb{Z}^+} \langle g, \phi_k\rangle K \phi_k(x) =  \sum_{k \in \mathbb{Z}^+} \langle g, \phi_k \rangle \lambda_k \phi_k(x),$$
here, $\phi_k$ are the eigenfunctions of the Koopman operator on $L^2.$ In the text, are the projections (the inner products) $c_k$ and the eigenfunctions $\phi_k$? If yes, $\phi_k$ should be a function of $x.$

2.  Why does the gradient projection accelerate optimization, given that the optimization dynamics is completely changing?

3. Are the different columns of the matrix $W_i$ time snapshots? In the DMD algorithm, they have to be.

3. What is the justification for projection on to the ``stable'' singular vectors? It is true that the eigenvalues close to the unit circle, but within the unit disk, have a correspondence with almost invariant structures or "slowly decaying modes", but these are not precisely used (even if qualitatively) in designing KGA. The slow decay is in the correlations between observables. If KGA did involve identifying the observables that show slow decay of autocorrelations for instance, that might more sense qualitatively.

4. In the numerical experiments, do we revert to SGD or some other optimization algorithm after a few epochs? What happens if we train for longer with KGA?

---

> ### Author Response · Authors · 2023-11-23
> **Response to Reviewer aC1Y (part 1/2)**
>
> Thank you for your insightful comments.
>
> > **Weakness 1** - Regarding the main concern on optimization guarantees:
> >
> We appreciate your concern regarding the convergence of our proposed method. To address this, we revisit the formulation of Koopman-based Gradient Acceleration (KGA).
>
> The standard Stochastic Gradient Descent (SGD) can be simplified as follows:
> $w_{i+1} = w_i - \eta \nabla$.
> In contrast, the proposed KGA is formulated as:
> $w_{i+1} = w_i - \eta \nabla - \alpha \left( U Q_{stable} \Lambda_{stable} L_{stable} U^T \eta \nabla \right) $
>
> From this comparison, it becomes evident that the convergence of our method is primarily guided by the original SGD dynamics. The additional component in KGA primarily contributes to accelerating specific directions. This type of formulation is widely used in optimization methods such as Momentum, Adam, etc.
>
> It's important to note that these components, which do not strictly converge to a zero gradient, can sometimes offer advantages, such as avoiding becoming stuck in local minima and saddle points. This characteristic is one of the reasons why such methods have gained popularity despite the lack of strict convergence guarantees in every scenario. We believe this perspective aligns well with current practices in the field and contributes to the robustness and effectiveness of our proposed method.
>
> > **Weakness 2** - Regarding usage of terminology:
> >
> We project the weight update onto the rank-r truncated left singular vectors of $W_i$. It’s in this space that we find a matrix approximation of the Koopman Operator ($\tilde{A}$). This follows the standard dynamic mode decomposition formulation (e.g., Schmid 2010). For the demarcation of ‘stable’, ‘unstable’, and ‘neutral’ Koopman modes, we follow the convention of Avila and Mezic from Nature Communications 2020 (https://doi.org/10.1038/s41467-020-15582-5). These are the terminologies defined to facilitate the reference and usage of the three categories of Koopman modes. We agree the usage of “stable”, “unstable”, and “neutral” might be unconventional, but it saves the trouble to have to use an entire sentence everytime we need to refer to these three groups of modes.
>
> We have removed the term ‘stable manifold’ from the paper. In this case, the equilibrium that the stable manifolds approach is “convergence” of a network but we understand that this is less rigorous than the traditional notion of a stable manifold.
>
> > **Weakness 3** - Regarding the usage of DMD and the proposed method changing dynamics:
> >
> The reviewer is absolutely right that DMD is the technique developed to “approximate” the infinite-dimensional Koopman operator with a finite-dimensional counterpart. There have indeed been many variants to standard DMD, including, for example, extended DMD (Williams et al. 2015)  and Hankel DMD (Arbabi & Mezic, 2017). The major challenge these variants attempt to resolve is to consider an extended basis of observables in order to provide better approximations of the Koopman operator. For example, Hankel DMD tries to lift the measurement data to a higher-dimensional space using delay embedding. In our study, however, since the observables are in the weight space, which is already very high dimensional (in the order of millions), the standard DMD is sufficient. In Sec. 2, we provided  a reference (Kutz et al., 2016) as a comprehensive discussion of DMD.
>
> The reviewer is also correct that the proposed KGA and KWM would both change the dynamics of the training process which is why we re-calculate the dynamics in each training epoch. We did compare the performance of re-calculating the dynamics each iteration (KGA-Iter) and each epoch (KGA-Epoch), as shown in Fig. 4. KGA-Iter shows the best improvement in loss, but incurs a significant performance penalty. We choose to use KGA-Epoch for the tradeoff between computational efficiency and improvement in training loss. Please refer to the first set of comparisons in Sec. 4.1.
>
> M. O. Williams, I. G. Kevrekidis, and C. W. Rowley. A data–driven approximation of the Koopman operator: Extending dynamic mode decomposition. *Journal of Nonlinear Science*, 25(6):1307–1346, 2015.
>
> H. Arbabi and I. Mezic. Ergodic theory, dynamic mode decomposition, and computation of spectral properties of the Koopman operator. *SIAM Journal on Applied Dynamical Systems*, 16(4):2096–2126, 2017.

---

> ### Author Response · Authors · 2023-11-23
> **Response to Reviewer aC1Y (part 2/2)**
>
> Response to questions:
>
> > **Question 1** - Regarding the equations for the Koopman mode decompositions
> >
> We have double-checked the notations in Eqs. 2 and 3, and Fig. 1 and they are consistent with references cited (Mezic, 2021)(Mezic, 2005). In a sense, both the reviewer and the authors are right! $C_k(x)$ is indeed the projection of $g(x)$ onto the eigenspace $\phi_k$ and thus can be written as $<g(x), \phi_k>$. $\phi_k$, the eigenfunction of the Koopman operator, is indeed a function of $x$. However, here, it has been proven that $\phi$ is invariant under the dynamics of $T$ and its level sets are invariant sets. Please refer to Eqs. (51) and (60) in (Mezic, 2021).
>
> > **Question 2** - Why does the gradient projection accelerate optimization, given that the optimization dynamics is completely changing?
> >
> The reviewer is correct that our method indeed changes the underlying optimization dynamics, which is why we need to recalculate the dynamic every iteration or every epoch. Please also refer to the response to Weakness 3 and re-stated here that “We did compare the performance of re-calculating the dynamics each iteration (KGA-Iter) and each epoch (KGA-Epoch), as shown in Fig. 4. KGA-Iter shows the best improvement in loss, but incurs a significant performance penalty. We choose to use KGA-Epoch for the tradeoff between computational efficiency and improvement in training loss. Please refer to the first set of comparisons in Sec. 4.1.”
>
> > **Question 3** - Are the different columns of the matrix  $W_i$ time snapshots?
> >
> Yes. Different columns of the $W$ matrix are the snapshot (or weights) of each iteration or epoch. Please see response to #2 on why we choose KGA-Epoch over KGA-Iter.
>
> > **Question 4** - What is the justification for projection on to the ``stable'' singular vectors?
> >
> We project onto the rank-r truncated singular vectors (the remaining singular vectors are assumed to explain noise). We then project to the stable left-eigenvectors of the linear operator in this rank-r subspace. By projecting onto the stable left-eigenvectors, we are getting the component of the weight that is slow-moving or “almost” invariant. The hypothesis is that the contribution from slowly decaying modes persists longer during standard SGD and that speeding up convergence of these slowly decaying modes will accelerate training.
>
> > **Question 5** - In the numerical experiments, do we revert to SGD or some other optimization algorithm after a few epochs? What happens if we train for longer with KGA?
> >
> KGA does not revert to SGD. We apologize for the confusion. In Sec. 4.1 (KGA vs. Prediction-based Koopman Acceleration), we were trying to compare the performance of KGA and the recent training acceleration approaches but based on Koopman’s predictive capability (Dogra & Redman, 2020)(Tano et al., 2020). We showed that although these methods perform well on smaller networks as the reference show, they would fail for relatively larger networks (even an FCN). The “bookstrap” approach refers to the usage of prediction-based acceleration with our tweak that when prediction starts introducing too much error, it reverts to Adam. We show performance increase, but still not as stable and good as the proposed KGA. We have clarified the writeup of that experiment. Please refer to Fig. 6 (a).

---

### Author Response · Authors · 2023-11-23
**Summary of Revisions**

We sincerely thank all the reviewers for their insightful reviews and valuable comments. We appreciate the opportunity to clarify and elaborate on the concerns raised. We have uploaded the revised manuscript. For responses to each question, please see the individual responses to each reviewer. In the following, we highlight the major changes:

1. Reorganize the paper to move more results from the Appendix to the main paper to highlight the performance gain vs. cost (runtime and memory usage) comparison of the proposed approach.

2. Conduct additional experiments of the proposed approach

      a.  in handling acceleration of networks with different types of loss functions.

      b.  in improving baseline optimization algorithms, including SGD, Adam, and Momentum (newly added).

      c.  with a comprehensive evaluation of runtime.

3. Clarify on the difference between  KGA and prediction-based Koopman acceleration.

---

### Meta-Review · Area_Chair_Qfd3 · 2023-12-05

**Metareview:**

This paper introduces a modified training algorithm based on a Koopman mode decomposition. After the discussion phase, the reviewers are not convinced by either the theoretical or empirical results. There was some concern about whether the heuristics used had sufficient motivation, and there are other questions about whether the performance improvements could justify the increased computational cost. The authors did update the paper with additional data and discussion about the computational cost, and while the concern is partially addressed, questions about the practical impact of the method remain.

**Justification For Why Not Higher Score:**

Lack of sufficiently strong theoretical motivation and empirical support for the proposed method.

**Justification For Why Not Lower Score:**

N/A

---

### Decision · Program_Chairs · 2024-01-16

Reject